# Macrovascular Networks on Contrast-Enhanced Magnetic Resonance Imaging Improves Survival Prediction in Newly Diagnosed Glioblastoma

**DOI:** 10.3390/cancers11010084

**Published:** 2019-01-14

**Authors:** Josep Puig, Carles Biarnés, Pepus Daunis-i-Estadella, Gerard Blasco, Alfredo Gimeno, Marco Essig, Carme Balaña, Angel Alberich-Bayarri, Ana Jimenez-Pastor, Eduardo Camacho, Santiago Thio-Henestrosa, Jaume Capellades, Javier Sanchez-Gonzalez, Marian Navas-Martí, Blanca Domenech-Ximenos, Sonia Del Barco, Montserrat Puigdemont, Carlos Leiva-Salinas, Max Wintermark, Kambiz Nael, Rajan Jain, Salvador Pedraza

**Affiliations:** 1Department of Radiology, University of Manitoba, Winnipeg, MB R3T 2N2, Canada; messig@exchange.hsc.mb.ca; 2Research Unit of Diagnostic Imaging Institute (IDI), Department of Radiology (Girona Biomedical Research Institute) IDIBGI, Hospital Universitari Dr Josep Trueta, 17007 Girona, Spain; carlesbiarnes90@gmail.com (C.B.); gbs.blasco@gmail.com (G.B.); fredigimeno1989@gmail.com (A.G.); marianmarti33@gmail.com (M.N.-M.); bl.domenech@gmail.com (B.D.-X.); sapedraza@gmail.com (S.P.); 3Department of Computer Science, Applied Mathematics and Statistics, University of Girona, 17003 Girona, Spain; pepus@imae.udg.edu (P.D.-i.-E.); santiago.thio@udg.edu (S.-T.H.); 4Medical Oncology, Institut Catala Oncologia (ICO), Applied Research Group in Oncology (B-ARGO), IGTP, Badalona, 08916 Barcelona, Spain; cbalana@iconcologia.net; 5QUIBIM SL, Quantitative Imaging Biomarkers in Medicine, 46026 Valencia, Spain; angel@quibim.com (A.A.-B.); anajimenez@quibim.com (A.J.-P.); educamacho@quibim.com (E.C.); 6Department of Radiology, Hospital del Mar, 08003 Barcelona, Spain; jaumecapellades@gmail.com; 7Philips Healthcare Ibérica, 28050 Madrid, Spain; javier.sanchez.gonzalez@philips.com; 8Medical Oncology, Institut Catala Oncologia (ICO), 17007 Girona, Spain; Sdelbarco@iconcologia.net; 9Hospital Cancer Registry, ICO, Hospital Universitari Dr Josep Trueta, 17007 Girona, Spain; mpuigdemont@iconcologia.net; 10Department of Radiology, University of Missouri, Columbia, MO 65212, USA; carlosleivasalinas@gmail.com; 11Department of Radiology, Neuroradiology Division, Stanford University, Stanford, CA 94304, USA; mwinterm@stanford.edu; 12Department of Radiology, Icahn School of Medicine at Mount Sinai, New York, NY 10029, USA; kambiznael@gmail.com; 13Departments of Radiology and Neurosurgery, New York University School of Medicine, New York, NY 10016, USA; Rajan.Jain@nyumc.org

**Keywords:** glioblastoma, magnetic resonance imaging, angiogenesis, biomarker, survival

## Abstract

A higher degree of angiogenesis is associated with shortened survival in glioblastoma. Feasible morphometric parameters for analyzing vascular networks in brain tumors in clinical practice are lacking. We investigated whether the macrovascular network classified by the number of vessel-like structures (nVS) visible on three-dimensional T1-weighted contrast–enhanced (3D-T1CE) magnetic resonance imaging (MRI) could improve survival prediction models for newly diagnosed glioblastoma based on clinical and other imaging features. Ninety-seven consecutive patients (62 men; mean age, 58 ± 15 years) with histologically proven glioblastoma underwent 1.5T-MRI, including anatomical, diffusion-weighted, dynamic susceptibility contrast perfusion, and 3D-T1CE sequences after 0.1 mmol/kg gadobutrol. We assessed nVS related to the tumor on 1-mm isovoxel 3D-T1CE images, and relative cerebral blood volume, relative cerebral flow volume (rCBF), delay mean time, and apparent diffusion coefficient in volumes of interest for contrast-enhancing lesion (CEL), non-CEL, and contralateral normal-appearing white matter. We also assessed Visually Accessible Rembrandt Images scoring system features. We used ROC curves to determine the cutoff for nVS and univariate and multivariate cox proportional hazards regression for overall survival. Prognostic factors were evaluated by Kaplan-Meier survival and ROC analyses. Lesions with nVS > 5 were classified as having highly developed macrovascular network; 58 (60.4%) tumors had highly developed macrovascular network. Patients with highly developed macrovascular network were older, had higher volume_CEL_, increased rCBF_CEL_, and poor survival; nVS correlated negatively with survival (*r* = −0.286; *p* = 0.008). On multivariate analysis, standard treatment, age at diagnosis, and macrovascular network best predicted survival at 1 year (AUC 0.901, 83.3% sensitivity, 93.3% specificity, 96.2% PPV, 73.7% NPV). Contrast-enhanced MRI macrovascular network improves survival prediction in newly diagnosed glioblastoma.

## 1. Introduction

Glioblastoma is the most common angiogenic malignant astrocytic tumor. Despite therapeutic advances, the prognosis of glioblastoma is dismal, with median overall survival of 16 months [1]. Angiogenesis, a key step in tumor progression, is among the most important prognostic factors in glioblastoma and correlates with worse survival [2,3,4,5,6,7,8]. However, morphological vascular parameters cannot easily differentiate between pre-existing brain vessels incorporated into tumors and neoangiogenesis [8], though tumor vascularity and leakiness measured with perfusion imaging have been shown to indirectly correlate with different stages of angiogenesis with increasing glioma grade [9]. Thus, it is more useful to assess vascularity by considering the tumor’s “vascular network”, which comprises of both pre-existing vessels incorporated into the tumor and microvessels arising from neoangiogenesis [10,11]. However, no morphometric parameter has been validated as a biomarker of vascularity. A biomarker that enabled tumor grading based on angiogenic sub-patterns could help improve diagnosis and prognosis, and would facilitate the translation of antiangiogenic therapy from the experimental stage into clinical practice. In addition to classic angiogenesis seen at histology as evenly distributed capillary-like microvascular sprouting, immunohistochemistry for CD34 reveals unevenly distributed bizarre vascular formations (glomeruloid vascular formations, vascular garlands, and vascular clusters), which are considered a histological hallmark of glioblastoma [3,5]. These formations are considered a late, secondary development that is insufficient to save tumoral tissue from hypoxia-mediated death. Birner et al. [3] found that a predominantly bizarre vascular pattern was associated with worse survival than a predominantly classic pattern. Moreover, histological microvessel density is higher in the classic vascular pattern than in the predominantly bizarre pattern.

However, various factors impede the use of histopathological or molecular biomarkers to assess glioblastoma angiogenesis in clinical practice. Continuous monitoring is difficult because sampling is invasive, does not represent the heterogeneity within tumors, requires long processing times as well as complex storage and testing techniques, and yields poor interobserver agreement [12]. By contrast, imaging biomarkers for patient selection and therapeutic monitoring are not limited by invasive sampling and can assess the heterogeneity of the whole tumor. Novel magnetic resonance imaging (MRI) biomarkers of angiogenesis of newly diagnosed glioblastomas have aroused great interest. The degree of vascularity of glioblastomas has been evaluated directly by digital subtraction angiography [13,14,15,16] and magnetic resonance angiography (MRA) [17], as well as indirectly by dynamic susceptibility contrast perfusion MRI (DSC-MRI) [9,15,16,18,19,20,21,22,23,24,25] and CT perfusion [26]. DSC-MRI shows that glioblastomas have regions of significantly elevated relative cerebral blood volume (rCBV) or relative cerebral blood flow (rCBF) consistent with their increased vascularity [18,19,20,21,22]. However, some newly diagnosed glioblastomas show low vascularity [14,15,16,17]. Wetzel et al. [16] documented the absence of vascularity in 30% of 231 gliomas, and a recent pilot study using gadofosveset blood-pool contrast agent found that half of glioblastomas had decreased vascularity on high-resolution MRA [17]; both these studies found that increased vascularity predicted worse outcome [16,17].

Numerous studies suggest that pretreatment MRI features may be prognostic indicators of survival in patients with glioblastoma [16,17,27,28,29,30,31,32,33]. Features included in survival models include the degree of necrosis, the degree of enhancement, multifocality, satellite lesions, volume of contrast-enhancing lesion (CEL), volume of non-CEL, and extent of edema. Additionally, a few studies have suggested that including imaging features based on Visually AcceSAble Rembrandt Images (VASARI), a controlled vocabulary system developed to standardize the grading of visual MRI features in gliomas [34], could enhance the predictive power of survival models based on clinical features [27,31]. Other studies have suggested that the extent of resection, age at diagnosis, and Karnofsky Performance Scale (KPS) score can determine survival in glioblastoma [32,35,36].

Using MRI biomarkers to assess baseline risk at diagnosis can help researchers stratify patients into risk groups and help clinicians make treatment decisions. This study aimed to assess whether including measures of the macrovascular network of newly diagnosed glioblastoma in routine clinical contrast-enhanced MRI protocols could improve the predictive power of survival models based on clinical and other imaging features. We counted vessel-like structures related to the CEL or non-CEL components on three-dimensional contrast-enhanced spin-echo T1-weighted imaging (3D-T1CE) to assess the macrovascular network. We found that the macrovascular network on 3D-T1CE, together with age at diagnosis and standard treatment, best predicted survival with AUC = 0.901 (*p* < 0.001), yielding 83.3% sensitivity, 93.3% specificity, 96.2% positive predictive value, and 73.7% negative predictive value.

## 2. Results

### 2.1. Determination of the Cutoff for Number of Vessel-Like Structures in Glioblastoma

The best cutoff for nVS related to the CEL or non-CEL components to discriminate between highly developed macrovascular network and less developed macrovascular network in newly diagnosed glioblastomas on 3D-T1CE (Figure 1), was 5, yielding 100% sensitivity, 97.7% specificity, 98.1% positive predictive value, and 100% negative predictive values. Considering glioblastomas with nVS > 5 to have highly developed macrovascular network, 53 (54.6%) tumors had highly developed macrovascular network.

Interobserver agreement for macrovascular network was excellent (*κ* = 0.85).

### 2.2. Patient Characteristics

Table 1 summarizes the patients’ data. All 97 patients (66 male, mean age 58 ± 15 years) died during the observation period. The median KPS score at diagnosis was 80.0 (IQR, 70.0–80.0).

Patients with glioblastomas with highly developed macrovascular network were older, had higher mean volume_CEL_ and higher rCBF_CEL_, and tended toward lower KPS scores (median, 85.76% vs. 90.20% in patients with glioblastomas with less developed macrovascular network, *p* = 0.063) (Table 1).

Standard treatment was administered in 64 (65.98%) patients, 53 with highly developed macrovascular network and 44 with less developed macrovascular network. None of the patients underwent antiangiogenic therapy.

Ependymal invasion was more common in lesions with highly developed macrovascular network (55% vs. 38%, *p* = 0.013). Interobserver agreement for macrovascular network was excellent (*κ* = 0.85). Interobserver agreements for the 13 features from the VASARI features were good to excellent; the highest agreement was for midline cross (*κ* = 0.957) and the lowest for eloquent area involvement (*κ* = 0.778).

### 2.3. Survival Analysis According the Treatment Received and Macrovascular Network

Figure 2 shows the Kaplan-Meier survival curves according to the macrovascular network and treatment received. In patients who received standard treatment, the survival was significantly longer (Figure 2A). For the subgroup of patients who received standard treatment, nVS was also negatively correlated with survival (*r* = −0.347; *p* = 0.016). Median survival rates for patients with less developed macrovascular network and patients with highly developed macrovascular network were 11.67 months (95% CI, 4.51–18.05) and 7.80 months (95% CI, 3.48–13.21), respectively (Figure 2B). Median survival rates for patients with less developed macrovascular network increased and patients with highly developed macrovascular network decreased receiving standard treatment were 15.9 months (95% CI, 11.24–20.70) and 10.26 months (95% CI, 8.03–18.26), respectively. When treatment was other, median survival for patients with less developed macrovascular network and patients with highly developed macrovascular network was 3.80 months (95%CI, 2.59–8.03) and 3.30 months (95% CI, 1.75–5.69), respectively (Figure 2C). For overall tumors, nVS was negatively correlated with survival (*r* = −0.286; *p* = 0.008) (Figure 2D).

Univariate Cox proportional hazards regression analysis showed that age at diagnosis, macrovascular network, delay mean time at CEL (DMT_CEL_), DMT_NCEL_, apparent diffusion coefficient at NCEL (ADC_NCEL_), ependymal invasion, thickness of CEL margin, and the treatment received are the variables that significantly discriminate the survival time (Table 2).

The cutoff values are reported in Table 3. Standard treatment was the best predictor of survival (hazard ratio: 0.163, 95% CI, 0.092–0.288; *p* = 0.001) with a sensitivity, specificity, positive predictive value and negative predictive value of 62.5%, 93.1%, 94.6%, 56.2%, respectively (AUC = 0.778) (Table 3). However, in the best multivariate model selection hazard ratios, the most important combined factors were age at diagnosis, standard treatment and macrovascular network with a sensitivity, specificity, positive predictive value and negative predictive value of 83.3%, 93.3%, 96.2%, 73.7%, respectively (AUC = 0.901, *p* < 0.001).

## 3. Discussion

This study using a routine clinical MRI protocol found that the macrovascular network of newly diagnosed glioblastoma could be classified based on nVS on standard postcontrast sequences and that macrovascular network is a prognostic biomarker of survival. Patients with highly developed macrovascular network had worse survival. Interestingly, the clinical and imaging characteristics of patients with the two subtypes of macrovascular network were very similar, the only difference being that glioblastomas with highly developed macrovascular network had higher volume_CEL_ and rCBF_CEL_ than those with less developed macrovascular network. These findings are consistent with the well-recognized observations that glioblastomas are highly vascularized tumors with highly heterogeneous angioarchitectures and that solid tumor growth depends on vascularity [3,4,11]. Although the vascular network initially develops by incorporating existing host vessels, solid tumors probably cannot grow more than 1 mm^3^ unless they synthesize their own new vessels [37].

Our results allow us to speculate that MRI might offer two approaches to assessing glioblastomas’ vascularity in the pretreatment stage: the macrovascular network and the microvascular network. In fact, DSC-MRI perfusion evaluates the dynamic process of the first-pass effect in the short period of time during which the contrast agent enters the brain parenchyma, providing physiological information regarding vascularity: rCBV indirectly reflects vascularity and is useful for grading gliomas, as well as a predictive and prognostic tool [38,39,40,41,42]. In all patients, compared with contralateral normal-appearing WM, DSC-MRI perfusion indices and permeability values were increased in both CEL and non-CEL, supporting the idea that newly diagnosed glioblastomas have a highly developed vascular network. Recently, Jia et al. [26] demonstrated that imaging permeability parameters correlated positively with microvascular density in glioblastomas. However, the possible role of microvascular density as a biomarker of survival is still controversial. Some studies have defended its value as predictor of glioma growth and survival [2,43,44], and another found that low-grade gliomas with higher microvascular density had a higher risk of malignant transformation and were associated with shorter survival [45]. By contrast, other studies found no significant correlation with survival [2,4]. These discrepancies may be related to the type of angiogenesis that is predominant in the tumor. Birner et al. [3] found that a prominent classical capillary pattern of angiogenesis with fewer bizarre glomeruloid vessels was an independent predictor of longer survival in patients with glioblastoma. In tumors with the bizarre pattern, cell growth seems to outpace neovascularization, so their progression seems less dependent on adequate vascular networks. Glioblastomas with a predominantly bizarre pattern are unlikely to benefit from antiangiogenic therapy. By contrast, in glioblastomas with a classic capillary-like pattern, cell growth seems to parallel neovascularization; the evenly distributed pattern of delicate vessels would make it easier for chemotherapeutic drugs to reach tumor cells, so patients are more likely to respond better to antiangiogenic therapy.

We assessed the vascular network in two ways. The first, DSC-MRI perfusion indices, clearly showed that all the glioblastomas in our patients had developed a microvascular network. The second, visually counting vessel-like structures related to the CEL or non-CEL on 3D-T1CE, detected an increased presence of large-diameter vessels (cutoff, nVS = 5) in only 60% of glioblastomas. The multivariate analysis found that nVS had a greater impact than other quantitative or qualitative imaging variables when added to age and standard treatment to predict survival in the multivariate analysis. Age and standard treatment were independent prognostic factors (AUC = 0.850), and adding macrovascular network increased the predictive power of the model (AUC = 0.901). Since glioblastomas need the vascular network to provide nutrients and oxygen for metabolism and removal of waste products, the nVS could reflect differences in vascular growth, remodeling related to blood flow and metabolic demands (glioblastomas with nVS > 5 had higher rCBF_CEL_ and higher volume_CEL_). Although ependymal invasion was more frequent in glioblastomas with highly developed macrovascular network, adding ependymal invasion in multivariate analysis did not increase the predictive power of the model. Nevertheless, recent studies show that glioblastomas with ependymal invasion are more frequently associated with multifocal lesions and tumor recurrence than those without and that this variable can also serve as an independent prognostic factor for poor survival [35,46,47,48]. We focused on variables available in the pretreatment stage because being able to predict survival at this stage can help clinicians determine the timing, mode, and aggressiveness of treatment; by contrast, variables related to surgery or postoperative treatments cannot be assessed until later in the course of disease and are not useful in initial treatment planning.

Glioblastoma stem cells stimulate tumor angiogenesis by expressing elevated levels of vascular endothelial growth factor (VEGF), and high levels of VEGF correlate with worse prognosis [49]. Bevacizumab (Avastin; Genentech, San Francisco, CA, USA), a monoclonal antibody that inhibits VEGF, results in tumor shrinkage in 55% of patients with recurrent glioblastomas, palliating neurological symptoms and increasing progression-free survival [50]. However, bevacizumab does not benefit all patients. Bevacizumab combined with temozolomide and radiotherapy failed to improve survival in some patients with newly diagnosed glioblastoma [51,52], and bevacizumab combined with lomustine did not increase overall survival over lomustine alone in patients with recurrent glioblastoma [53]. Therefore, there is an urgent need for new biomarkers to predict the response to antiangiogenic treatment so responders can be selected before initiating treatment. Along these lines, rCBV and rCBF maps have been used to predict survival in patients under antiangiogenic therapy [54,55]. Our results suggest that macrovascular network would be a helpful biomarker to identify high-risk patients who would likely benefit from more aggressive and experimental treatments, and promises to be helpful in predicting survival under antiangiogenic therapy and in guiding subsequent treatment.

These results are a step toward validating the results of our recent pilot study [17], where we described these patterns of vascularity on gadofosveset trisodium high-resolution MRA in a small sample of 35 newly diagnosed glioblastomas [17]. Gadofosveset trisodium (MS-325; in that study, Vasovist^®^; Bayer Schering Pharma AG, Berlin, Germany; now commercially available as Ablavar^®^, Lantheus Medical Imaging, North Bilerica, MA, USA) is a blood-pool contrast agent that binds reversibly to albumin, resulting in much higher blood relaxivity and better contrast-to-noise ratio of the vessels than non-protein binding extracellular contrast agents, such as gadobutrol. Gadofosveset trisodium has a half-life about 16 h with optimal enhancement and diagnostic information at 6 h after administration [56]. Nevertheless, gadofosveset trisodium is indicated for MRA only to evaluate aortoiliac occlusive disease, and it is not included in clinical protocols to study brain tumors. Thus, in the present study, we used intravenous gadobutrol (Gadovist^®^; Bayer Schering Pharma, Berlin, Germany), an extracellular contrast agent whose high relaxivity and high concentration yields the highest T1 shortening time and excellent image enhancement. Extensive routine clinical experience has demonstrated that gadobutrol is effective and well tolerated [57].

Various limitations of this study merit comment. This was a single-center study, with a modest sample size. Thus, further studies with larger populations are necessary to determine the usefulness of nVS in predicting survival in patients with glioblastoma. We quantified the number of vessels without postprocessing. This approach is easy and fast, but also subjective; nevertheless, the interobserver agreement was excellent. Although our cutoff of nVS = 5 discriminated well between highly vascularized and less vascularized glioblastomas well, further studies are needed to explore the performance of other possible cutoffs. Moreover, interactions between factors such as the field strength of equipment, type of gadolinium-based contrast agents, and MRI acquisition protocols present challenges to establishing macrovascular network as a biomarker for predicting survival, because differences across institutions impede the replication of single-center study results [38].

Glioblastoma’s macrovascular network phenotype may result from unusual levels or combinations of angiogenic factors that lead to unbalanced angiogenesis. Bennett et al. [58] reported that circulating endothelial cell counts were increased in patients with glioblastoma, and that preoperative circulating progenitor cell counts appeared to be proportional to tumor vascularity, as they correlated strongly with rCBV_CEL_. Further studies correlating with histological, molecular techniques, and angiogenic gene expression might help validate different glioblastoma subtypes according to their macrovascular network. Likewise, macrovascular network would be a MRI biomarker signature that may identify distinct GBM phenotypes associated with highly significant survival differences and specific molecular pathways. It seems imaging could detect different angioarchitectural patterns of glioblastoma (not included in the present analysis) (Figure 1 and Figure 3) in a manner similar to histopathology [3,10]. In low-grade gliomas, rounder vessel contour and microvascular branching may reflect increased intraluminal pressure due to disorderly vasculature [59]. This qualitative approach should be followed by quantitative morphometric analyses. Thus, future studies could also assess other vessel-related parameters, such as size, shape, density, distribution, and branching patterns to better express the glioblastoma’s complex angioarchitecture [60,61]. High- and ultra-high field MRI might enable better analysis and quantification of vessel morphology [62,63]. Computer-assisted fractal analysis is potentially useful for quantifying different vascular networks and identifying different vascular patterns [64]. Standard pathology tests are part of clinical practice and essential criteria for most clinical trials, so a direct comparison between the imaging vascular networks and histological assessments (e.g., blood-vessel immunohistochemistry for blood vessels) must be performed in the field to adopt an additional method to complement or replace the histological analysis

## 4. Material and Methods

### 4.1. Study Data

We prospectively recruited consecutive patients aged > 18 years with newly diagnosed, histopathologically confirmed glioblastoma. We excluded patients with a history of other malignancies. Between April 2014 and November 2016, we considered 115 patients; 18 patients were excluded because histology ruled out glioblastoma. Therefore, 97 patients (62 men; age, 58 ± 15 years) were included in the analysis. Our institution’s ethics committee approved the study protocol, and all patients provided written informed consent. This Ethic Committee does not provide a code because in that institution clinical trials have a code only. This is not a clinical trial, this an academic study. The date when we obtained the approval was on 7 March, 2014. Patients were managed according to published guidelines [42], recommending surgical resection followed by radiotherapy combined with concomitant and adjuvant temozolomide chemotherapy as standard treatment. Survival was defined as the interval between the date of initial glioblastoma diagnosis and the date of either death from any cause or the last follow-up date on which the patient was known to be alive.

### 4.2. MRI Protocol

Patients received no corticosteroids before MRI. Patients underwent MRI on a standard clinical 1.5-T system (Intera, Philips Healthcare, Best, The Netherlands) with an eight-channel head coil. The protocol included axial spin-echo T1-weighted imaging, fluid-attenuated inversion recovery imaging (FLAIR), single-shot echo-planar diffusion-weighted imaging (DWI), and first-pass perfusion DSC-MRI with gadobutrol (Gadovist; Bayer Schering Pharma, Berlin, Germany), and 3D-T1CE. Parameters for axial T1-weighted spin-echo imaging were repetition time (TR) 536 ms, echo time (TE) 15 ms; flip angle, 90°; matrix, 256 × 192; section thickness, 5 mm; intersection gap, 1 mm; field of view (FOV), 230 × 180 mm; and brain coverage, 120 mm. Parameters for FLAIR were TR, 7569 ms; TE, 115 ms; inversion delay, 2200 ms; flip angle, 90°; matrix, 256 × 192; section thickness, 5 mm; intersection gap, 1 mm; (FOV), 230 × 180 mm; and brain coverage, 120 mm. For DWI, parameters were TR, 3758 ms; TE, 99 ms (*b* = 0 and 1000 s/mm^2^); 20 sections; section thickness, 5 mm; intersection gap, 1 mm; FOV, 230 × 230 mm; and matrix, 192 × 128; two signals were acquired in three orthogonal directions and combined into a trace image. Apparent diffusion coefficient (ADC) maps were calculated on a voxel-by-voxel basis. For perfusion DSC-MRI, multislice T2* single-shot echo-planar images were acquired before, during, and after rapid administration of a contrast bolus (sixteen 7-mm sections without gap; matrix, 128 × 128; FOV, 230 × 230 mm; TR, 1800 ms; TE, 25 ms; flip angle, 90°). The TR of each multi-shot block was 17 ms, and the acquisition time for each dynamic volume was 1.8 s. TE was 17 ms and the flip angle 7°. Each perfusion series consisted of 50 dynamic acquisitions with temporal resolution set to 1.8 s during the first pass of a standard dose (0.1 mmol/kg bolus of gadobutrol injected at 5 mL/s with a power injector, followed by a 20-mL bolus of saline at the same rate). To reduce the effect of contrast leakage on CBV calculations, a 5 mL bolus of gadobutrol was administered 5 min prior to DSC-MRI perfusion acquisition, at a rate of 1 mL/s, followed by a 15 mL saline flush. Parameters for isovoxel 3D-T1CE images were TR/TE, 536/15 ms; flip angle, 90°; matrix, 256 × 192; and section thickness, 1 mm.

### 4.3. Quantitative Image Analysis

Digital imaging and communications in medicine files were transferred to an external computing station for processing. Using Olea Sphere V.3.0 software (Olea Medical; La Ciotat, France), we applied a semi-automatic region-growing segmentation algorithm on selected seeds to delimit volumes of interest (VOI) of CEL, necrosis, and non-CEL. Non-CEL was defined as the hyperintense area on FLAIR surrounding the CEL that was associated with mass effect and architectural distortion, including blurring of the gray matter/white matter (WM) interface [43]. We also obtained the VOI in normal-appearing WM of the centrum semiovale of the hemisphere contralateral to the tumor from three contiguous slices manually. To create DSC-MRI perfusion parametric maps with contrast-leakage correction, we applied a fully automated deconvolution analysis of the tissue concentration-versus-time curve with arterial input function following the technique outlined by Boxerman et al. [43]. All VOIs were co-registered onto the DSC-MRI parametric maps (Figure 3). The following parameters were calculated: rCBF, rCBV, mean transit time (MTT), delay mean time (DMT), time when the residue function reaches its maximum (Tmax), time to peak (TTP), and ADC.

### 4.4. Qualitative Image Analysis

For each patient, two neuroradiologists (J.P. and S.P., with 13 and 25 years’ experience, respectively) blinded to clinical data independently reviewed axial T1-weighted images, 3D-T1CE, and axial FLAIR images on a local picture archiving and communication system (Starviewer, Gilab, University of Girona; Girona, Spain), scoring 13 features according to VASARI (tumor location; side of lesion center; eloquent area involvement; multifocality; satellite lesion; enhancement quality; thickness of CEL margin; deep WM invasion; midline cross; hemorrhage; ependymal invasion; pial invasion; and mass effect). Eloquent area involvement was defined as the presence of tumor in the cortex or immediate subcortical WM of speech-motor or speech-receptive brain areas [45]. Multifocality was defined as the presence of at least one region of tumor not contiguous with the dominant lesion, outside the region of signal abnormality surrounding the dominant mass. A satellite lesion was defined as an area of enhancement within the region of signal abnormality surrounding the dominant lesion, but not contiguous in any part with the dominant lesion mass. Deep WM invasion was defined as the presence of CEL or non-CEL in the internal capsule, corpus callosum, or brain stem. Ependymal involvement was defined as CEL in contact with the lining of the ventricles. Interobserver agreement for the MRI features was assessed, and discordant interpretations were resolved by consensus. To assess the macrovascular network, readers counted vessel-like structures related to the CEL or non-CEL components on 3D-T1CE (Figure 1), analyzing source images and multiplanar reconstructions together with subvolume maximum-intensity projection slabs. To achieve optimal views of the vessels of interest, readers were free to manipulate these images by changing subvolume position and thickness interactively in real time. Before the final analysis, readers reached a consensus on discordant classifications. Assessing the macrovascular network took less than four min per patient.

### 4.5. Statistical Analysis

Continuous variables are expressed as means and standard deviations or medians and ranges; categorical variables are expressed as frequencies and percentages. We used Student’s *t*-tests to determine differences in continuous variables when normal distribution was able to be assumed or Wilcoxon test when normality was not able to be assumed, and chi-square tests to determine differences in categorical variables with respect to the macrovascular network and survival. To calculate the cutoff point for nVS to discriminate between glioblastomas with high nVS and those with low nVS (i.e., those with a highly developed macrovascular network vs. those with a less developed macrovascular network), we used receiver operating characteristic (ROC) curves. To determine independent predictors of overall survival, we used univariate Cox proportional hazards regression, selecting variables with *p*-values < 0.05 to generate prognostic models and calculating hazard ratios with corresponding 95% confidence intervals. Then, we developed multivariate Cox proportional hazards models from clinical and MRI findings including age at diagnosis, macrovascular network, DMT_CEL_, DMT_NCEL_, nVS, ependymal invasion, thickness of CEL, and treatment (dichotomized “standard treatment” or “others”), to adjust for the influence of these prognostic factors. We also used ROC analysis to determine the optimal cutoff for these variables for predicting survival. We defined the optimal cutoff value as the value that maximizes the sum of sensitivity and specificity. Survival curves were calculated by the Kaplan-Meier method, using the variables most significant in differentiating between survival for more than one year versus less than one year. We also combined these variables to achieve the best predictive capability, using the log-rank test to analyze overall differences. To assess interobserver agreement for the imaging features, we used the kappa consistency test, considering *k* > 0.81 excellent agreement, 0.61 < *k* < 0.80 good agreement, and *k* < 0.60 poor agreement. Statistical analyses were performed with R version 3.4.2 (R Foundation for Statistical Computing, Vienna, Austria) and IBM SPSS version 23.0.0.0 (IBM Corp. Armonk, NY, USA). Significance was set at *p* < 0.05 for all tests.

## 5. Conclusions

In conclusion, the macrovascular network of newly diagnosed glioblastoma on contrast-enhanced MRI promises to be an easily assessable biomarker of survival. Cross-validation studies in other populations are necessary to test the generalizability of our findings, to expand our understanding of the pathophysiology of glioblastoma’s vascular network, and to determine whether this approach can effectively improve subpopulation selection before treatment. Clinical trials need to clarify the possible influence of vascular patterns on the effects of antiangiogenic therapies. In addition to helping clinicians plan patients’ management, the macrovascular network, together with age and standard treatment may also be useful for stratifying glioblastoma patients for enrollment in clinical trials.

## Figures and Tables

**Figure 1 cancers-11-00084-f001:**
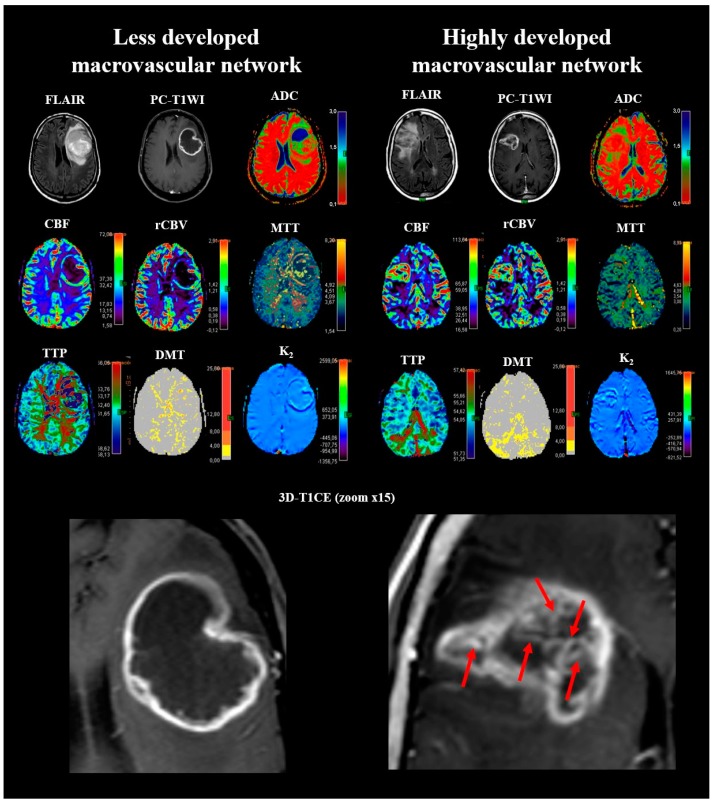
Macrovascular network on 3D-T1 contrast enhancement MRI. Panel of 2 cases illustrating MRI characteristics for less developed and highly developed macrovascular networks. The glioblastoma with a highly developed macrovascular network (red arrows) shows an unevenly distributed bizarre large-vessel pattern. ADC indicates apparent diffusion coefficient; DMT, delay mean time; FLAIR, fluid-attenuated inversion recovery; K_2_, microvascular permeability; MTT, mean transit time; rCBF, relative cerebral blood flow; rCBV, relative cerebral blood volume; PC-T1WI, postcontrast T1 weighted image; TTP, time to peak.

**Figure 2 cancers-11-00084-f002:**
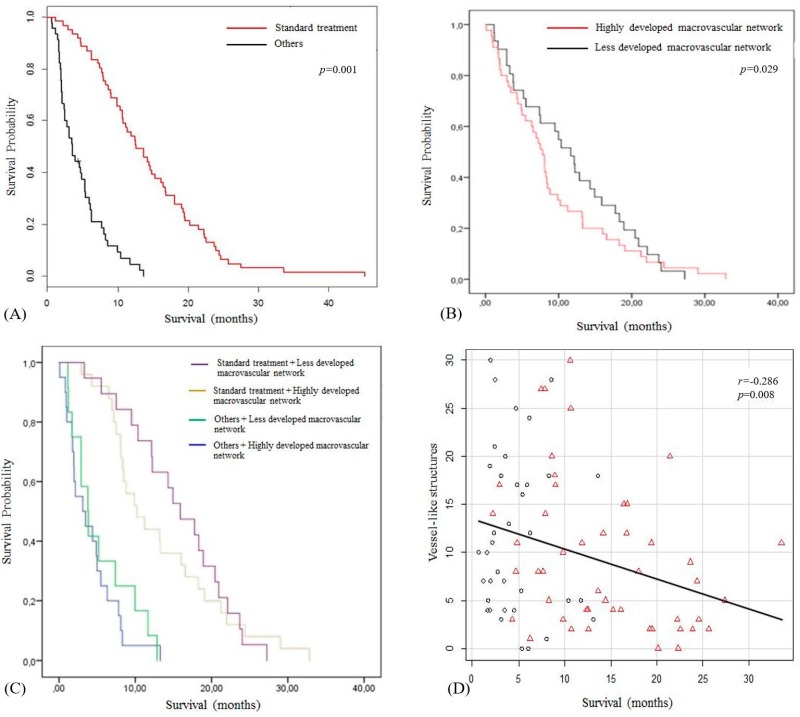
Survival analysis. Kaplan-Meier survival curves comparing survival rates according to treatment received (**A**) between macrovascular network subtypes on contrast-enhanced MRI (**B**), and 4 branches of groups combining treatment received and macrovascular network on contrast-enhanced MRI (**C**). Patients with glioblastomas with less developed macrovascular networks that received standard treatment had better survival. The number of visible vessels significantly correlates with survival (**D**).

**Figure 3 cancers-11-00084-f003:**
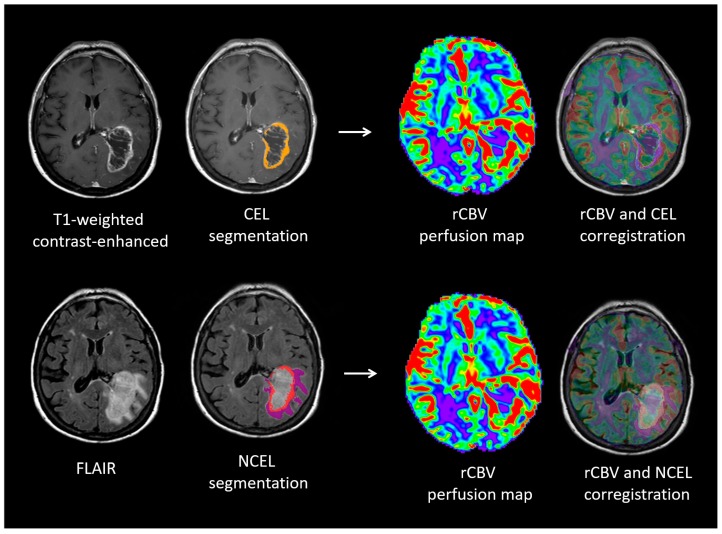
Lesion segmentation and co-registration MRI analysis. Regions of interest for contrast-enhancing lesion and non-contrast enhancing lesion in a mass consistent with glioblastoma involving the left temporal and parietal lobes. DSC-MRI perfusion color maps demonstrating the areas of increased relative cerebral blood volume (red) within the tumor. Postcontrast imaging shows an increased macrovascular network consisting of an evenly distributed, more linear vessel pattern within the necrotic component of the lesion (upper left).

**Table 1 cancers-11-00084-t001:** Patient characteristics.

Characteristic	Overall (*n* = 97)	Less Developed Macrovascular Network (*n* = 44)	Highly Developed Macrovascular Network (*n* = 53)	*p*-Value
Male (%)	62 (63.9%)	66 (68.2%)	58 (60.4%)	0.426
Age at diagnosis (years)	58 (15)	54 (15)	61 (12)	0.026
Karnofsky score	87.45 (18.23)	90.20 (7.65)	85.76 (14.02)	0.063
Volume_CEL_ (cm^3^)	20.4 (19.12)	13.55 (10.52)	26.28 (22.71)	0.001
Volume_NCEL_ (cm^3^)	50.67 (32.64)	44.91 (27.6)	55.62 (36.02)	0.260
Volume of necrosis (cm^3^)	20.37 (21.84)	21.52 (28.2)	19.41 (14.86)	0.310
rCBF_CEL_	16.08 (4.92)	14.31 (4.06)	18.52 (5.04)	0.001
rCBF_NCEL_	16.45 (4.34)	15.5 (4.05)	17.75 (4.49)	0.118
rCBV_CEL_	1.76 (0.93)	1.66 (1.09)	1.83 (0.81)	0.181
rCBV_NCEL_	2.08 (1.18)	1.88 (0.86)	2.22 (1.37)	0.361
MTT_CEL_ (s)	5.74 (1.93)	5.71 (1.68)	5.77 (2.12)	0.938
MTT_NCEL_ (s)	6.06 (2.23)	6.14 (2.57)	6 (2)	0.898
TTP_CEL_ (s)	25.99 (8.67)	25.86 (9.26)	26.09 (8.38)	0.667
TTP_NCEL_ (s)	25.85 (7.12)	24.67 (5.67)	26.64 (7.96)	0.296
DMT_CEL_ (s)	−0.26 (1.21)	−0.38 (1.35)	−0.18 (1.11)	0.573
DMT_NCEL_ (s)	−0.12 (0.57)	−0.16 (0.51)	−0.1 (0.62)	0.725
Microvascular permeability, K_2CEL_	−56.48 (57.21)	−51.75 (67.45)	−59.9 (49.48)	0.274
Microvascular permeability, K_2NCEL_	−53.82 (61.23)	−46.25 (83.83)	−59.31 (38.29)	0.099
ADC_CEL_ (mm^2^ s^−1^ × 10^−3^)	0.30 (0.07)	0.30 (0.07)	0.31 (0.08)	0.952
ADC_NCEL_ (mm^2^ s^−1^ × 10^−3^)	0.44 (0.03)	0.44 (0.03)	0.44 (0.03)	0.944
Vessel-like structures (*n*)	9.56 (8.11)	2.75 (1.94)	15.21 (6.83)	<0.001
Treatment				0.280
Surgery + RT + TMZ (%)	64 (65.98%)	33 (34.01%)	31 (31.96%)	
Surgery + RT (%)	11 (11.34%)	5 (5.14%)	6 (6.19%)	
RT + TMZ (%)	14 (14.42%)	3 (3.08%)	11 (11.34%)	
TMZ (%)	7 (7.22%)	3 (3.08%)	4 (4.11%)	
Palliative (%)	1 (1.03%)	0 (0%)	1 (1.03%)	

Note—Unless otherwise specified, data are means with standard deviations in parentheses. Significant differences between less developed macrovascular network and highly developed macrovascular network cohorts are highlighted in bold. ADC indicates apparent diffusion coefficient; CEL, contrast-enhancing lesion; DMT, delay mean time; MTT, mean transit time; NCEL, non-CEL; rCBF, relative cerebral blood flow; rCBV, relative cerebral blood volume; RT, radiotherapy; TMZ, temozolamide; TTP, time to peak.2.1. Sub-types of macrovascular networks on contrast-enhanced MRI.

**Table 2 cancers-11-00084-t002:** Associations between variables and survival time.

Variable	Overall (*n* = 97)	Less than 1 Year (*n* = 56)	More than 1 Year (*n* = 29)	*p*-Value
Male (%)	62 (63.9%)	23 (79.3%)	33 (58.9%)	0.060
Age at diagnosis (years)	57.75 (14.43)	62.07 (12.66)	50 (15.11)	<0.001
Karnofsky score	87.45 (18.23)	82.52 (13.15)	91.03 (12.66)	0.084
Highly/less developed macrovascular network	44/53	16/40	18/11	0.002
CEL (cm^3^)	20.4 (19.12)	22.41 (21.16)	19.36 (17.04)	0.421
Non-CEL (cm^3^)	50.67 (32.64)	48.33 (27.55)	50.73 (40.82)	1.000
Necrosis (cm^3^)	20.37 (21.84)	17.99 (15.45)	23.1 (22.48)	0.647
rCBF_CEL_	16.08 (4.92)	15.95 (4.74)	15.59 (5.39)	0.512
rCBF_NCEL_	16.45 (4.34)	16.69 (3.46)	15.33 (5.94)	0.463
rCBV_CEL_	1.27 (0.73)	1.42 (0.88)	1.05 (0.36)	0.342
rCBV_NCEL_	1.54 (0.95)	1.64 (1.09)	1.42 (0.73)	0.763
MTT_CEL_ (s)	5.74 (1.93)	5.86 (1.79)	5.59 (2.29)	0.424
MTT_NCEL_ (s)	6.06 (2.23)	5.79 (1.91)	6.31 (2.79)	0.485
TTP_CEL_ (s)	25.99 (8.67)	24.55 (5.91)	28.78 (13.08)	0.590
TTP_NCEL_ (s)	25.85 (7.12)	25.15 (6.13)	27.02 (9.75)	0.808
DMT_CEL_ (s)	−0.26 (1.21)	−0.58 (1.2)	0.46 (1.09)	0.006
DMT_NCEL_ (s)	−0.12 (0.57)	−0.26 (0.58)	0.18 (0.47)	0.010
Microvascular permeability, K_2 CEL_	−56.48 (57.21)	−64.46 (68.17)	−52.38 (26.57)	0.730
Microvascular permeability, K_2 NCEL_	−53.82 (61.23)	−58.03 (68.31)	−48.27 (55.32)	0.730
ADC_CEL_ (mm^2^ s^−1^ × 10^−3^)	0.3 (0.07)	0.3 (0.06)	0.31 (0.08)	0.831
ADC_NCEL_ (mm^2^ s^−1^ × 10^−3^)	0.44 (0.03)	0.43 (0.03)	0.45 (0.02)	0.041
Vessel-like structures (*n*)	9.56 (8.11)	12.46 (8.62)	6.59 (5.48)	0.002
Treatment				0.003
Standard treatment (%)	64 (65.98%)	28 (28.87%)	27 (27.84%)	
Surgery + RT (%)	11 (11.34%)	8 (8.25%)	1 (1.03%)	
RT + TMZ (%)	14 (14.43%)	12 (12.36%)	1 (1.03%)	
TMZ (%)	7 (7.22%)	7 (7.22%)	0 (0%)	
Palliative (%)	1 (1.03%)	1 (1.03%)	0 (0%)	

Note—Data are represented as means (standard deviations). Significant associations are highlighted in bold. ADC indicates apparent diffusion coefficient; CEL, contrast-enhancing lesion; DMT, delay mean time; MTT, mean transit time; NCEL, non-CEL; rCBF, relative cerebral blood flow; rCBV, relative cerebral blood volume; RT, radiotherapy; TMZ, temozolamide; TTP, time to peak.

**Table 3 cancers-11-00084-t003:** Survival prediction: Summary of class performance.

Variable	Area under Curve	Cutoff	Sensitivity	Specificity	Positive Predictive Value	Negative Predictive Value	Hazard Ratio (95% CI)	*p*-Value	Likelihood *p* (Multivariate)
**Univariate Analysis**									
Age at diagnosis	0.737	59.73	0.679	0.724	0.826	0.538	1.042 (1.022,1.063)	<0.001	
DMT_NCEL_	0.697	−0.50	0.267	1.000	1.000	0.405	0.444 (0.232,0.852)	0.015	
Vessel-like structures	0.709	6.94	0.696	0.621	0.780	0.514	1.029 (0.998,1.061)	0.033	
Highly developed macrovascular network	0.667	Present	0.714	0.621	0.784	0.529	1.254 (0.788,1.998)	0.029	
Standard treatment	0.778	Present	0.625	0.931	0.946	0.562	0.163 (0.092,0.288)	<0.001	
**Bivariate Analysis**									
Age at diagnosis DMT_NCEL_	0.859	58 −0.48	0.867	0.733	0.867	0.733	1.042 (1.014–1.071) 0.560 (0.284–1.105)	0.002 0.095	<0.001
Age at diagnosis Standard treatment	0.850	54.8 2.0	0.714	0.897	0.930	0.619	1.026 (1.005–1.048) 0.213 (0.117–0.388)	<0.001 0.015	<0.001
Vessel-like structures Standard treatment	0.864	5 Present	0.768	0.897	0.935	0.667	1.017 (0.987–1.048) 0.170 (0.096–0.301)	0.044 <0.001	<0.001
Highly developed macrovascular network Standard treatment	0.850	-	0.625	0.931	0.946	0.562	1.265 (0.792–2.019) 0.163 (0.092–0.288)	0.032 <0.001	<0.001
**Trivariate Analysis ***									
Age at diagnosis Standard treatment Highly developed macrovascular network	0.901	-	0.833	0.933	0.962	0.737	0.604 (0.459–0.796) 0.163 (0.090–0.297) 1.481 (0.909–2.414)	<0.001 <0.001 0.045	<0.001

* This trivariate analysis provided the highest performance of the model. Other combinations of variables with lower performance included age at diagnosis, standard treatment and vessel-like structures with a sensitivity, specificity, positive predictive value and negative predictive value of 0.801, 0.908, 0.925, 0.716, respectively (AUC = 0.863, *p* = 0.02), and age at diagnosis, standard treatment and DMT_NCE_ with a sensitivity, specificity, positive predictive value and negative predictive value of 0.793, 0.891, 0.901, 0.705, respectively (AUC = 0.824, *p* = 0.03).

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
