# Peer review of "Macrovascular Networks on Contrast-Enhanced Magnetic Resonance Imaging Improves Survival Prediction in Newly Diagnosed Glioblastoma"

_cancers, 2019, doi:10.3390/cancers11010084_

Round 1
Reviewer 1 Report
In this paper the Authors propose the macrovascular network classified by the number of vessel- like structures (nVS) as a a prognostic biomarker of survival in newly diagnosed glioblastomas. GBM is deadly disease and the standard of care only increases in few months the life of patients..There is an urgent need for new biomarkers to predict the response to antiangiogenic treatment so responders can be selected before initiating treatment; therefore the paper is interesting. However, there are two major caveats in the study
The Authors performed multiple comparison but the resulting p values were adjusted neither using Benjamin –Hochberg (B_H) procedure nor Bonferroni’s rule. Why?
They used Student’s_t-test, a parametric test, to determine differences in continuous variables: did all parameters analysed follow a gaussian distribution? Why the Authors did not use a non parametric test?
Previous studies showed that for glioblastoma subsets can be discriminated on genetic fingerprints , such as IDH1/IDH2 or MGMT hypermethylation . No information on molecular features of GBMs included in the study is reported.
Minor points
“RT + TMZ” Does it mean that patients performed only biopsy?
“Patients received no corticosteroids before MRI”. It seems unusual, how many days before MRI did they interrupt the treatment?
“Interestingly, the clinical and imaging characteristics of patients with the two subtypes of macrovascular network were very similar”, It is not clear to me
Were not patients with highly developed macrovascular network older ? Did not they perform more often only biopsy?
Author Response
Comments to reviewers
We greatly appreciate all the positive comments and valuable suggestions from both reviewers. We have revised the manuscript “Macrovascular networks on contrast-enhanced magnetic resonance imaging improves survival prediction in newly diagnosed glioblastoma’, based on the reviewer’s comments. Having addressed all of the comments, we feel that our manuscript is very much improved. We would like to respond to the reviewers’ comments as below.
All changes in the manuscript are marked in red.
# Reviewer 1
The Authors performed multiple comparison but the resulting p values were adjusted neither using Benjamin–Hochberg (B_H) procedure nor Bonferroni’s rule. Why?
We thank the reviewer for this remark. We show a selection of the best univariate survival predictors and over these selected predictors we present only the significant bivariante and trivariate models. All this models, except one, are with a p-value less than 0.001 and we have not considered applying adjustment methods like Benjamnin-Hochberg or Bonferroni.
They used Student’s_t-test, a parametric test, to determine differences in continuous variables: did all parameters analysed follow a gaussian distribution? Why the Authors did not use a non parametric test?
In fact, not all parameters follow a Gaussian distribution. We applied a t-test for Gaussian distributed data and the equivalent non parametric test, Wilcoxon test, for non-Gaussian data. Moreover, due to the sizes of two groups, both tests provide equivalent results, in most cases the difference is in the third decimal value. In Statistical Analysis, we replaced ‘We used Student’s t-tests to determine differences in continuous variables and chi-square tests to determine differences in categorical variables with respect to the macrovascular network and survival’ for ‘We used Student’s t-tests to determine differences in continuous variables when normal distribution was able to be assumed or Wilcoxon test when normality was not able to be assumed, and chi-square tests to determine differences in categorical variables with respect to the macrovascular network and survival’.
Previous studies showed that for glioblastoma subsets can be discriminated on genetic fingerprints, such as IDH1/IDH2 or MGMT hypermethylation. No information on molecular features of GBMs included in the study is reported.
Thank you for this comment. The molecular fingerprint/signature is lacking this paper. This is the next step that we want to develop. Extensive multiplatform genomic characterization has provided a higher-resolution picture of the molecular alterations. These studies provide the emerging view that GBM represents several histologically similar yet molecularly heterogeneous diseases, which influences taxonomic classification systems, survival, and therapeutic decisions. In light of the results of our paper, we believe with a high degree of probability some gene signature could be related with GBM macrovascular network. Invasive tumors are shown to have a different genomic composition and metabolic abnormalities that allow for a more aggressive GBM phenotype and resistance to therapy. Here, we discriminate two types of lesions and probably may constitute distinct disease subtypes developing through different genetic pathways. We are currently analyzing the genetic and molecular profile in other 2 larger samples of GBM (GLIOCAT sample and TCGA). Imaging-genomic analyses may prove invaluable in detecting novel targetable genomic pathways. We reflected this point in the Discussion (lines 264-266) as ‘Further studies correlating with histological, molecular techniques, and angiogenic gene expression might help validate different glioblastoma subtypes according to their macrovascular network’. To emphasize this concept, we have added ‘Likewise, macrovascular network would be a MRI biomarker signature that may identify distinct GBM phenotypes associated with highly significant survival differences and specific molecular pathways’.
Minor points
“RT + TMZ” Does it mean that patients performed only biopsy?
Currently, the treatment of newly diagnosed GBM is highly depending on surgical resection, radiotherapy, and chemotherapy. However, these patients did not accept surgery and we offered radiotherapy plus chemotherapy to them. We know that the efficiency of chemotherapy, given as the adjunct to RT or before RT, is still controversial. However, several systematic reviews have been carried out to provide reliable evidences for the chemotherapy combined with RT in newly diagnosed GBM (Yin AA, Zhang LH, Cheng JX, et al. Radiotherapy plus concurrent or sequential temozolomide for glioblastoma in the elderly: a meta-analysis. PloS One 2013;8:e74242; and Stewart LA. Chemotherapy in adult high-grade glioma: a systematic review and meta-analysis of individual patient data from 12 randomised trials. Lancet 2002; 359:1011-8).
“Patients received no corticosteroids before MRI”. It seems unusual, how many days before MRI did they interrupt the treatment?
Thanks to comment this. The exact mechanisms by which the corticosteroids produce its therapeutic action in patients with intracranial tumors still remain unclear. Its principal effect is to reduce edema formation, hence lowering intracranial pressure. This edema resolution is achieved by either decreasing the blood tumor barrier permeability to small solutes or increasing parenchymal resistance to fluid transport. Dexamethasone may act directly on the cerebral vasculature. Several previous imaging studies have investigated whether dexamethasone affects cerebral perfusion in intracranial tumors. Unfortunately, the results of these studies are contradictory possibly because of methodologic differences and heterogeneous tumor patient populations. In our study, patients received no corticosteroids before MRI in order to avoid some potential effects of these drugs on diffusion or hemodynamic perfusion imaging parameters. According to the clinic concerned, the patients were studied with computed tomography in emergency setting before MRI or underwent MRI directly. Our department where this study has been carry out allows to do this logistic circuit due to the proximity of imaging modalities. When we suspect GBM on CT the patients underwent MRI protocol in the next 9 hours (mean, 6.5 hours) after providing written informed consent. Bastin and colleagues suggested that dexamethasone does not significantly affect tumor blood flow but may, by reducing peritumoral water content and local tissue pressure, subtly increase perfusion in the edematous brain (non-contrast enhancing lesion). After 48–72 hours of corticosteroid treatment, tumor (contrast enhancing lesion) cerebral blood flow (CBF) was practically unchanged, whereas edematous CBF increased on average by 11.6%. The view that dexamethasone produces no change in tumor perfusion and an increase in peritumoral edema is contradicted by other studies. Leenders et al found that dexamethasone produced a significant reduction in both CBF and CBV in tumor and contralateral tissue, but not in peritumoral edema, 1–5 days after treatment. Behrens et al reported a 32% decrease in peritumoral edema CBF compared with contralateral white matter in 11 patients with malignant glioma treated with dexamethasone (12–24 mg/day) for a least 6 days by using Xe-CT. Østergaard and colleagues measured changes in CBF and CBV relative to contralateral white matter and blood-tumor barrier permeability 1-6 hours after corticosteroid therapy in 3 patients with astrocytoma, 2 with oligodendroglioma, and one with CNS lymphoma. They reported a dramatic decrease in blood-tumor barrier permeability, a significant 15% reduction in rCBV for peritumoral gray matter (directly adjacent to peritumoral edema). Except for the patient with lymphoma, they found that both peritumoral gray and white matter rCBF did not show any systematic change after steroid treatment. Animal models have shown that the corticosteroids can act as an indirect anti-angiogenic agent by decreasing expression of vascular endothelial growth factor. Badruddoja et al. demonstrated an anti-angiogenic effect using MRI-derived cerebral blood volume maps to non-invasively evaluate changes in 9L tumor vasculature following dexamethasone treatment. Dexamethasone had a significantly lower mean CBV and mean vessel diameter than untreated ones indicating an inhibition of vascular growth. In our study, we aimed to avoid any effect of corticosteroids on the contrast and non-contrast enhancing lesion and this was the reason to acquire the MRI protocol before corticosteroids administration.
“Interestingly, the clinical and imaging characteristics of patients with the two subtypes of macrovascular network were very similar”, It is not clear to me.
We wanted to remark that the only difference was focused on volume and cerebral blood flood of contrast-enhancing lesion (both parameters higher in lesions with highly developed macrovascular network). Volume of non-contrast enhancing lesion, volume of necrosis, and blood volume, mean transit time, time-to-peak, delay mean time, ADC and permeability (K2) of contrast enhancing lesion were similar in both types of GBMs. There were no differences for all parameters of non-contrast enhancing lesion component.
Were not patients with highly developed macrovascular network older? Did not they perform more often only biopsy?
In fact, patients with highly developed macrovascular network were older than patients with less developed macrovascular network (61 ± 12 vs 54 ± 15 years). Age impacts directly to the survival as well. However, in terms of biopsy with a cut off at 67 years (this was the best cutoff in terms of survival prediction in our sample) there was no statistical significance (p=0.456).
Reviewer 2 Report
This is an interesting manuscript in line with scope of “Cancers”.
The aim of the study was to investigate if diagnostic imaging modalities could assist treating clinicians better define patient prognosis.
In principle the article is worthy of publication however the explanation of their data is insufficient, and justification of their conclusion is not immediately obvious. Thus I would recommend significant rewriting of the manuscript to clarify the results.
I would like to clarify that I have expertise in non-invasive imaging and glioblastoma, but I am not a statistics expert. Therefore it was a challenge for me to understand the results tables presented. I invite the editors to consider their typical reader demographics when taking this review into account.
Specific comments:
The paper appears to have been written for a format where the methods come before the results; however, in this version the methods follow the discussion. I recommend the paper be rewritten considering the format for Cancers, and possibly including many of the figures from the methods section in the results section, as much of this information is required before showing the data in the tables.
Abstract/Introduction
Minor issues:
· By definition all tumours have angiogenesis, so I recommend rephrasing the first sentence in the abstract for accuracy: “Angiogenesis is associated with shortened survival in glioblastoma”.
Methodology
Major issues:
· The paper is written with the expectation the methods are read before the results section. Since they are not in this order the paper needs to be rewritten to enable the reader to understand the results.
Results
Major issues:
· It was challenging to decipher and understand the tables in this paper as there was very little commentary to direct the attention of the reader.
· It is not clear why the authors compare survival as a variable with survival time in table 2. These data presented in the last 7 rows is confusing. I assume that this is a breakdown of survival in months in patients in different groups but this is not described and I can’t understand why some boxes are empty.
· table 2 needs reformatting as it is not clear what the p value for “treatment” represents – is this data of all different treatments combined? it seems obvious that whether or not the patient received treatment is a predictor of outcome.
· The descriptions for table 3 require improvement for the reader to understand the data. There is no explanation regarding why multiple hazard ratios are reported, nor how the authors defined “cutoff”
Minor issues:
· Labels in table 3 are not in the same format as other tables or the text (ie. subscripted text)
· Figure 3 comes before figures 1 & 2
· Table 3, correct univariate, bivariate
· The fact that table 3 is spread over two pages makes it extremely hard to read and needs reformatting
Discussion/Conclusions
Major issues:
· There is only one trivariate analysis presented, however there is no evidence that other combinations of three variables are inferior to the one presented. Thus it is hard to see how the authors arrived at the conclusion that this three variable model is the best.
· It is not clear if the additional effort required to perform the MRI analysis adds significant value to the patient or treating clinician.
· It would be important to comment on whether the MRI analysis adds anything significant over traditional histology/pathology tests to the patient or treating clinician. Standard pathology tests will always be a part of clinical practice or essential criteria for most clinical trials, so to convince those in the field to adopt an additional method, or replace these histological assessments (eg. IHC for blood vessels), a direct comparison between the different modalities should be made.
Minor issues:
· The discussion is lengthy and could be more concise without compromising the paper’s quality
Author Response
Comments to reviewers
We greatly appreciate all the positive comments and valuable suggestions from both reviewers. We have revised the manuscript “Macrovascular networks on contrast-enhanced magnetic resonance imaging improves survival prediction in newly diagnosed glioblastoma’, based on the reviewer’s comments. Having addressed all of the comments, we feel that our manuscript is very much improved. We would like to respond to the reviewers’ comments as below.
All changes in the manuscript are marked in red.
# Reviewer 2
Abstract/Introduction
Minor issues:
By definition all tumours have angiogenesis, so I recommend rephrasing the first sentence in the abstract for accuracy: “Angiogenesis is associated with shortened survival in glioblastoma”.
Thanks for this important detail. We correct as ‘Higher degree of angiogenesis is associated with shortened survival in glioblastoma’.
Methodology
Major issues:
The paper is written with the expectation the methods are read before the results section. Since they are not in this order the paper needs to be rewritten to enable the reader to understand the results.
We really appreciate this comment. We have reorganized the information in order to improve this issue. The information in the tables have been improve detailing the parameters better. Footnotes have been inserted in the tables. Importantly, Figure 1 have been moved in this section.
Results
Major issues:
It was challenging to decipher and understand the tables in this paper as there was very little commentary to direct the attention of the reader.
We found in the tables the way to provide the maximum amount of information resulting from our analysis. The recent approach to study GBM in two components, contrast and non-contrast enhancing lesion, significantly extended this data. We only try to report in text format the main results of these analyses. In line with the last query, we think that the footnotes inserted in the tables will help the reader.
It is not clear why the authors compare survival as a variable with survival time in table 2. These data presented in the last 7 rows is confusing. I assume that this is a breakdown of survival in months in patients in different groups but this is not described and I can’t understand why some boxes are empty.
The reviewer is right. In order to clarify the data provided in the table 2, we have decided to delete the last 7 rows. This information is provided by Figure 2 where the survival curves are illustrated in combination with the 4 arms of treatment according the macrovascular network.
Table 2 needs reformatting as it is not clear what the p value for “treatment” represents – is this data of all different treatments combined? it seems obvious that whether or not the patient received treatment is a predictor of outcome.
In the last lines of the Table 2 we try to provide the percentage of patients respect to the treatment received. The significant p-value represent the contrast for all 5 different types of clinical management.
The descriptions for table 3 require improvement for the reader to understand the data. There is no explanation regarding why multiple hazard ratios are reported, nor how the authors defined “cutoff”
To calculate the cutoff point for nVS to discriminate between glioblastomas with high nVS and those with low nVS (i.e., those with a highly developed macrovascular network vs. those with a less developed macrovascular network), we used receiver operating characteristic (ROC) curves. We have added to the text the following: ‘We defined the optimal cutoff value as value that maximizes the sum of sensitivity and specificity’.
Minor issues:
Labels in table 3 are not in the same format as other tables or the text (ie. subscripted text).
All labels in the table 3 have been written in the same format as other tables.
Figure 3 comes before figures 1 & 2
The order of the figures has been modified as the result of the text modification.
Table 3, correct univariate, bivariate
This lapses have been corrected.
The fact that table 3 is spread over two pages makes it extremely hard to read and needs reformatting
In agreement with the reviewer, we have deleted the non-significant p-values of the univariate analysis and the lowest AUC of the bivariate analysis.
Discussion/Conclusion
Major issues:
There is only one trivariate analysis presented, however there is no evidence that other combinations of three variables are inferior to the one presented. Thus it is hard to see how the authors arrived at the conclusion that this three variable model is the best.
We did an extensive work in combining all significant variables in threes in order to perform the trivariate analysis and check if this model provided higher AUC value. In fact, the combination of age at diagnosis, standard treatment and highly developed macrovascular network were the variables that in the trivariate model provided an AUC over 0.900. Other combinations of variables did not provide better performance and we decided not include this information in the table. We included this footnote in the Table 3 as follows: ‘*This trivariate analysis provided the highest performance of model. Other combinations of variables with lower performance were not included in the table’.
It is not clear if the additional effort required to perform the MRI analysis adds significant value to the patient or treating clinician.
We note that the macrovascular network of newly diagnosed GBM derived from clinical standard protocol (requiring no additional time) promises to be an easily assessable biomarker of survival. This information complements the value of the treatment. In fact, to perform the complete standard treatment (surgical resection followed by radiotherapy combined with concomitant and adjuvant temozolomide chemotherapy) is one the best predictors of survival. We also remark that cross-validation studies in other populations are necessary to test the generalizability of our results. We strongly believe that expanding the understanding of the pathophysiology of GBM’s vascular network (and possibly other vascularized tumors) may be useful for stratifying patients for enrollment in clinical trials. These points are highlighted in the conclusion of this paper.
It would be important to comment on whether the MRI analysis adds anything significant over traditional histology/pathology tests to the patient or treating clinician. Standard pathology tests will always be a part of clinical practice or essential criteria for most clinical trials, so to convince those in the field to adopt an additional method, or replace these histological assessments (eg. IHC for blood vessels), a direct comparison between the different modalities should be made.
This is a good remark that we include in the last paragraph of the Discussion. We have added ‘Standard pathology tests are part of clinical practice and essential criteria for most clinical trials, so a direct comparison between the imaging vascular networks and histological assessments (eg. blood-vessel immunohistochemistry for blood vessels) must be performed in the field to adopt an additional method to complement or replace the histological analysis’.
Minor issues:
The discussion is lengthy and could be more concise without compromising the paper’s quality.
We agree with the reviewer. However, these results are new in the literature and put them into the context to be followed by the readers do not allow us to shorten the discussion. The journal does not place limitation in word counting. We expressly asked this reviewer to consider the length of the discussion appropriate.
Round 2
Reviewer 1 Report
The manuscript is clearer and the Authors answered my questions.
Author Response
The manuscript is clearer and the Authors answered my questions.
Thank you for all your comments of the first revision that improve the quality of the paper.
Reviewer 2 Report
This review is for a revised manuscript
The paper has been marginally improved, however there remain deficiencies in the quality and descriptions in the paper. The authors have taken some feedback on board and tried to correct the flow of the paper, however it is my opinion that the paper really needs to be rewritten properly, as there is still too much relevant information included in the methods at the end of the paper. If the results were rewritten this would greatly improve the paper. The results are presented with no commentary to guide the reader through the data. This compromises the full understanding of the research and results in the reader being confused. Alternatively the authors might want to consider a different journal format where the methods are read before the results section.
Major issues:
1) Table 1 – there is no leading description in the body of the text describing the differences/criteria defining tumours with a “Less developed macrovascular network” or “Highly developed macrovascular network”. There is no description of which columns the p values in the table are comparing (overall vs les developed or highly developed?).
2) Table 1 – there are 5 different treatments listed, but only one p value. It is not clear what was compared. Same for survival – exactly which groups were compared?
3) Section 2.1 – no context for this information has been provided. Some of this text would make more sense if it was presented before table 1. I think it would read better to have figure 1 before table 1 so the readers can see how the authors have discriminated between less and high macrovascular networks before presenting all the comparisons
4) Section 2.2 – there is no attempt in the text to direct the reader to which parts of the figure the data relates (ie, parts a, b, c or d of figure 2)
Minor issues:
· The subheadings could be improved to be more descriptive
· Line 152 – the edited sentence doesn’t make sense “When treatment was other different from standard,”
· Table 2 – instead of simply stating that the best trivariate analysis is presented, I recommend including the rest of the comparisons as a supplementary figure to give more weight to the results.
Author Response
Comments to reviewer
We greatly appreciate all the positive comments and valuable suggestions from this reviewer. Again, we have revised the manuscript “Macrovascular networks on contrast-enhanced magnetic resonance imaging improves survival prediction in newly diagnosed glioblastoma’, based on the reviewer’s comments. Having addressed all of the comments, we feel that our manuscript is very much improved. All changes in the manuscript are marked in blue.
Major issues:
1) Table 1 – there is no leading description in the body of the text describing the differences/criteria defining tumours with a “Less developed macrovascular network” or “Highly developed macrovascular network”. There is no description of which columns the p values in the table are comparing (overall vs les developed or highly developed?).
In order to guide the reader through the data, we have added the following sentence before the Results section: ‘We counted vessel-like structures related to the CEL or non-CEL components on three-dimensional contrast-enhanced spin-echo T1-weighted imaging (3D-T1CE) to assess the macrovascular network’.
At the bottom of the table we have indicated ‘Significant differences between less developed macrovascular network and highly developed macrovascular network cohorts are highlighted in bold’ to better understand that the contrast is between both cohorts according to the macrovascular network.
2) Table 1 – there are 5 different treatments listed, but only one p value. It is not clear what was compared. Same for survival – exactly which groups were compared?
We provide the percentage of patients respect to the treatment received. The p-value represent the contrast for all 5 different types of clinical management. There were no differences in treatment received according the macrovascular network.
Regarding survival information, we have decided to delete these rows at the end of the Table 1. This information is explained into the body text and more developed in the rest of tables and figures.
3) Section 2.1 – no context for this information has been provided. Some of this text would make more sense if it was presented before table 1. I think it would read better to have figure 1 before table 1 so the readers can see how the authors have discriminated between less and high macrovascular networks before presenting all the comparisons.
The reviewer is right. We have reorganized the text better. We have introduced the information about the vessels cutoff and the figure where we illustrated the concept of macrovascular network BEFORE the Table 1. Now the text is more comprehensible.
4) Section 2.2 – there is no attempt in the text to direct the reader to which parts of the figure the data relates (ie, parts a, b, c or d of figure 2)
We have reorganized this sections as follows:
Figure 2 shows the Kaplan-Meier survival curves according to the macrovascular network and treatment received. Patients who received standard treatment the survival was significantly longer (Figure 2A). For the subgroup of patients who received standard treatment, nVS was also negatively correlated with survival (R=-0.347; P=0.016). Median survival rates for patients with less developed macrovascular network and patients with highly developed macrovascular network were 11.67 months (95% CI, 4.51-18.05) and 7.80 months (95% CI, 3.48-13.21), respectively (Figure 2B). Median survival rates for patients with less developed macrovascular network increased and patients with highly developed macrovascular network decreased receiving standard treatment were 15.9 months (95% CI, 11.24-20.70) and 10.26 months (95% CI, 8.03-18.26), respectively. When treatment was other different from standard, median survival for patients with less developed macrovascular network and patients with highly developed macrovascular network was 3.80 months (95%CI, 2.59-8.03) and 3.30 months (95%CI, 1.75-5.69), respectively (Figure 2C). For overall tumors, nVS was negatively correlated with survival (R=-0.286; P=0.008) (Figure 2D).
Minor issues:
· The subheadings could be improved to be more descriptive.
We have improved the subheadings as follows:
2.1. Determination of the cutoff for number of vessel-like structures in glioblastoma
2.2. Patient characteristics
2.3. Survival analysis according the treatment received and macrovascular network
· Line 152 – the edited sentence doesn’t make sense “When treatment was other different from standard,”
We simplified this as ‘When treatment was other’.
· Table 2 – instead of simply stating that the best trivariate analysis is presented, I recommend including the rest of the comparisons as a supplementary figure to give more weight to the results.
At the bottom of the Table 3 we have included ‘Other combinations of variables with lower performance included age at diagnosis, standard treatment and vessel-like structures with a sensitivity, specificity, positive predictive value and negative predictive value of 0.801, 0.908, 0.925, 0.716, respectively (AUC=0.863, P=0.02), and age at diagnosis, standard treatment and DMTNCE with a sensitivity, specificity, positive predictive value and negative predictive value of 0.793, 0.891, 0.901, 0.705, respectively (AUC=0.824, P=0.03)’.
Round 3
Reviewer 2 Report
The authors have addressed my comments and the manuscript has been improved.